Noise pollution has limited effects on nocturnal vigilance in peahens

Yorzinski Jessica L. jyorzinski@tamu.edu jly8@cornell.edu 1
Hermann Fredrick S. 2
1 Department of Wildlife and Fisheries Sciences, Texas A&M University , College Station , TX , United States
2 Department of Animal Sciences, Purdue University , West Lafayette , IN , United States
Kramer Donald
Electronic publication date: 2016 Sep 29
Publication date: 2016
Volume: 4
Electronic Location ID: e2525
Received 2016 Jul 19; Accepted 2016 Sep 4
Copyright: ©2016 Yorzinski and Hermann
Copyright year: 2016
Copyright holder: Yorzinski and Hermann
License: This is an open access article distributed under the terms of the Creative Commons Attribution License, which permits unrestricted use, distribution, reproduction and adaptation in any medium and for any purpose provided that it is properly attributed. For attribution, the original author(s), title, publication source (PeerJ) and either DOI or URL of the article must be cited.
License URL: https://creativecommons.org/licenses/by/4.0/

Keywords: Antipredator behavior, Pavo cristatus, Predator–prey, Sensory ecology, Noise pollution, Vigilance

Funding: College of Agriculture and Life Sciences at Texas A&M University Texas A&M AgriLife Research This research was funded by the College of Agriculture and Life Sciences at Texas A&M University and Texas A&M AgriLife Research. The funders had no role in study design, data collection and analysis, decision to publish, or preparation of the manuscript.

==============================
Natural environments are increasingly exposed to high levels of noise pollution. Noise pollution can alter the behavior of animals but we know little about its effects on antipredator behavior. We therefore investigated the impact of noise pollution on vigilance behavior and roost selection in an avian species, peafowl (Pavo cristatus), that inhabits urban environments. Captive peahens were exposed to noise pollution at night and their vigilance levels and roost selections were monitored. The vigilance levels of peahens were unaffected by exposure to noise pollution within trials. Furthermore, the peahens exhibited no preference for roosting farther or closer to noise pollution. Interestingly, predators often avoided the experimental area during nights with noise pollution, which could explain why vigilance rates were higher overall during control compared to noise trials. The results suggest that peahens’ perception of risk is not drastically impacted by noise pollution but longer-term studies will be necessary to assess any chronic effects.

Introduction

Noise pollution is increasingly prevalent in natural environments. Over 85% of the contiguous United States is exposed to noise pollution (Mennitt et al., 2013). Noise pollution is usually louder and more frequent than natural sounds in the environment and can therefore mask these natural sounds (Kight & Swaddle, 2011). In addition, noise pollution is often associated with other types of disturbances, such as light or chemical pollution (Halfwerk & Slabbekoorn, 2015). Because many animals use acoustic information to inform their behavioral decisions (Bradbury & Vehrencamp, 1998), noise pollution can have major impacts on their fitness (Patricelli & Blickley, 2006; Shannon et al., 2015).

Noise pollution could impact fitness because of its effect within a variety of different contexts. It can affect the mating behavior of animals. Pair bonds in zebra finches (Taeniopygia guttata) weaken when the birds are exposed to noise pollution (Swaddle & Page, 2007) and male sage grouse (Centrocercus urophasianus) attendance on breeding grounds decreases with noise pollution (Blickley, Blackwood & Patricelli, 2012). Noise pollution can impact territorial behavior as well. Many passerines adjust their songs, which function in both territory defense and mate attraction, to compensate for increased noise levels (Mockford & Marshall, 2009). They vocalize louder (Brumm , 2004), repeat songs (Brumm & Slater, 2006), or sing during times of low noise (Fuller, Warren & Gaston, 2007). Noise pollution can also impact parental investment; female house sparrows (Passer domesticus) provide less food to their young when living in noisy environments (Schroeder et al., 2012).

Less is known about the effects of noise pollution on antipredator behavior (Meillère, Brischoux & Angelier, 2015). However, noise pollution has the potential to alter animals’ perception of the environment (Quinn et al., 2006; Shannon et al., 2014a). For example, animals that are exposed to noise pollution may perceive the environment as more dangerous because their ability to detect auditory signals and cues is low (Quinn et al., 2006). Chaffinches (Fringilla coelebs) are more vigilant and peck less in response to noise pollution, suggesting that they perceive their noisy environment as risky. In contrast, other animals’ perception of their environment may not be impacted by noise pollution (Bejder et al., 2009). These animals could have already habituated to the noise if they never experienced negative effects in noisy environments or if they are generally tolerant of noise. For example, noise pollution does not impact gerbils’ (Gerbillus allenbyi and G. pyramidum) selection of a safe versus risky microhabitat (Abramsky et al., 1996) and therefore does not seem to alter their perception of the riskiness of their environment. Finally, some animals may perceive environments with noise pollution as relatively safe because noise-sensitive predators may avoid these areas (Francis, Ortega & Cruz, 2009). Elk (Cervus elephus) are less vigilant in response to noise pollution (Shannon et al., 2014a; Shannon et al., 2014b), suggesting that they view noisy environments as less risky. While we are beginning to understand how species perceive noise pollution with respect to risk levels during their active periods (daytime for diurnal species), we are unaware of any studies that have investigated this topic during their inactive periods (nighttime for diurnal species). Given that animals often rely on their senses differently depending on whether it is daytime or nighttime (e.g., some birds use their visual capabilities to forage during the day but switch to tactile capacities at night; Robert & McNeil, 1988), animals’ responses to noise pollution could vary depending on this factor.

We therefore examined the impact of noise pollution on nocturnal vigilance and roosting behavior in a diurnal avian species, peafowl (Pavo cristatus), that inhabits urban environments (Ramesh & McGowan, 2009). Peafowl are native to the Indian subcontinent and have also been introduced to other continents (Kannan & James, 1998). While they naturally live in deciduous forests and scrubby woodlands, they also live near human settlements (Ali & Ripley, 1969; Johnsingh & Murali, 1978). They roost atop tall structures at night (such as trees; De Silva, Santiapillai & Dissanayake, 1996 and are subject to predation by mammalian and avian predators De Silva, Santiapillai & Dissanayake, 1996; Kannan & James, 1998). There are three alternative explanations for how peafowl perceive noisy environments. First, if peafowl perceive their environments as risky when there is noise pollution, then we expect them to exhibit high rates of vigilance in response to noise pollution and avoid roosting in areas with high levels of noise pollution. Second, if peafowl are tolerant of noise pollution or have previously habituated to it, then we do not expect their vigilance rates to change from baseline levels and expect them to select their roosts irrespective of noise pollution. Third, if peafowl perceive environments with noise pollution as an escape from predators, then we expect their vigilance levels to decrease in response to noise pollution and we expect them to roost near high levels of noise pollution. In addition, because noise pollution can be stressful to some species and cause physiological changes (Blickley et al., 2012), we examined whether the mass of peahens changed depending on their exposure to noise pollution.

Methods

We explored the effect of artificial noise pollution on vigilance levels and roost selection in captive peahens at the Purdue Wildlife Area in West Lafayette, IN, USA (40.450327°N, −87.052574°E). The vigilance levels experiment was conducted between July 2014 and April 2015 and the roost experiment was conducted between March and June 2016. The experiments were performed in an outdoor experimental cage (4.5 m × 9.0 m) that was 75 m from the outdoor aviary (24.4 × 18.3 × 1.8 m) where the birds were permanently housed (the distance between the experimental cage and main aviary ensured that birds in the main aviary did not hear the broadcast noise from the experimental cage). The aviary was over 550 m from the nearest major road and it was surrounded by trees; noise pollution from anthropogenic sources was therefore minimal. The peahens were adults and were given food and water ad libitum. The study was approved by Purdue University Animal Care and Use Committee (#1305000862 & 1504001232).

Vigilance levels

We tested whether artificial noise pollution impacts vigilance levels in peahens (n = 30). For each trial, a peahen was put inside of the experimental cage for seven consecutive nights (the experimental design was similar to Yorzinski et al., 2015). The experimental cage had one wooden roost (0.85 m tall and 1.3 m long) that was 1.5 m from a rock-shaped speaker (150 W Outdoor Rock Speaker, model tfs6sl; TIC Corporation, City of Industry, California, USA). The speaker was connected to an audio amplifier (Audioengine N22; Austin, Texas) and an iPod touch (model A1509; Apple Corporation, Cupertino, California, USA). During noise trials, the speaker continuously broadcast white noise (white Gaussian noise generated with Matlab; 16 bit; 44.1 Hz) during nights (and days) 2–5; no noise was broadcast on nights 1, 6, or 7. The white noise automatically turned on at noon on the second day and turned off at noon on the sixth day (Woods outlet timer, model 50002, Mississauga, Canada). During medium noise trials (n = 10), the white noise had a sound pressure level (SPL) of 75 dB (A weighting; slow setting) at the middle of the roost; during loud noise trials (n = 10), the white noise measured 90 dB SPL at the middle of the roost (model 407730; Extech Instruments, Waltham, MA, USA). During control trials (n = 10), white noise was not broadcast on any of the seven nights. The medium and loud noise trials broadcast noise at the same decibel levels as used in a previous study on noise pollution and birds (Swaddle & Page, 2007); noise pollution in urban environments can exceed the decibel level that we broadcast in our loud noise trials (Chepesiuk, 2005). Furthermore, the peahens were not necessarily exposed to the same level of noise pollution for the entire trial. During the day, they could move to the opposite side of the experimental cage and therefore reduce the loudness of the noise that they experienced. Also, even though the peahens could have slept on the ground at the opposite side of the speaker at night, they always slept on the roost near the speaker.

The head movements of the peahens were continuously monitored with a 3-axis accelerometer (TechnoSmart, Rome, Italy; 3 mm × 1.1 mm; 0.5 g; sample resolution: 19.6 m s−2; sample rate: 50 Hz). The accelerometer was attached to a velcro strip (3.5 mm × 1.8 mm) that was glued (Artiglio Super 620) to the feathers atop the birds’ head (see Yorzinski et al., 2015 for further details on accelerometer and attachment). The accelerometer was replaced every day because the battery would not last for the duration of an entire trial. The accelerometer does not impact head movement rates (Yorzinski et al., 2015). We used a custom algorithm to extract the number of head movements the peahens exhibited during each night of the trial (starting 1 h after sunset and ending 1 h before sunrise; “nighttime period”); the accuracy of this algorithm is high (over 90% of head movements are correctly classified as head movements; see Yorzinski et al., 2015 for more details).

The peahens were weighed at the start and end of the trial (ZIEIS Veterinary Pet Scale; Apple Valley, Minnesota; 5 g accuracy). The length of the peahens’ tarsus + metatarsus was measured at the start of the entire experiment (Neiko digital caliper; Neiko Tools, Wenzhou, Zhejiang, China; model number: 01409 A; ± 0.03 mm accuracy). Three video cameras (Swann Pro-500; Swann Communications, Santa Fe Springs, California, USA) connected to a DVR (Swann DVR4-2600) recorded the experimental cage and the area immediately outside the experimental cage (2 m from the cage perimeter). Using these video recordings, we determined the amount of time that predators (raccoons and domestic cats) and non-predators (mice, frogs, flying squirrels, deer, and rabbits) were visible during the nighttime period. We also assessed the time at which the peahens ascended to and descended from the roost each night. The time at which a bird ascended to the roost for the night was assessed by moving backwards in the videos from the nighttime period (1 h after sunset) and finding the time when the bird jumped on the roost. If the bird was not already on the roost 1 h after sunset, then we moved forward in the videos until the bird jumped on the roost. The time at which a bird descended from the roost for the night was assessed in a similar way except that we moved forward in the videos from the nighttime period (1 h before sunrise) until finding the time when the bird jumped off the roost. If the bird was already off the roost 1 h before sunrise, we moved backward in the videos until the bird jumped off the roost. We excluded times when the experimenters interfered with when the bird ascended to the roost or descended from the roost.

Roost selection

In the first roost selection experiment, we tested whether peahens’ (n = 20) selection of nocturnal roosting locations was impacted by a medium-level of artificial noise pollution. For each trial, a peahen was put inside of the experimental cage (at least 4.5 h before sunset) for one night. The cage had two wooden roosts (0.85 m tall and 1.3 m long; 5.7 m between roosts) and a rock-shaped speaker (150 W Outdoor Rock Speaker, model tfs6sl; TIC Corporation, City of Industry, California, USA) that was positioned in front of each roost (1.4 m between the speaker and roost). One of the speakers (randomly selected for each trial) was connected to an audio amplifier (Audioengine N22; Austin, Texas, USA) and an iPod touch (model A1509; Apple Corporation, Cupertino, California, USA) that continuously broadcast white noise (white Gaussian noise generated with Matlab; 16 bit; 44.1 Hz). In the middle of the roost that was closer to the speaker, the white noise measured 75 dB SPL; in the middle of the roost that was farther from the speaker, the white noise measured 50 dB SPL (model 407730; Extech Instruments, Waltham, MA, USA). Two video cameras (Swann Pro-500) connected to a DVR (Swann DVR4-2600) recorded the experimental cage. Based on the video recordings, we determined whether the peahen slept on the roost closer or farther from the noise.

In the second roost selection experiment, we tested whether peahens’ (n = 20) selection of nocturnal roosting locations was impacted by a high-level of artificial noise pollution. The experimental procedure was the same as in the first roost selection experiment except the noise level was increased. In the middle of the roost that was closer to the speaker, the white noise measured 90 dB SPL; in the middle of the roost that was farther from the speaker, the white noise measured 65 dB SPL (Extech Instruments; model 407730). Due to a limited number of peahens, we tested 8 peahens that had not been used in the first roost selection experiment. In addition, we randomly selected 12 birds that we used in the first roost selection experiment and used them in this second experiment as well (at least 33 days lapsed since a bird was used in the first roost selection experiment; mean ± SE: 61 ± 6.1 d).

Data analysis

We tested whether artificial noise pollution impacts nocturnal vigilance rates. We ran a repeated-measures mixed linear model (PROC Mixed with a variance components covariance structure and the between-within degrees of freedom approximation) to examine whether vigilance rates differed among trials. The dependent variable was the natural log of the head movement rate (number of head movements during nighttime period divided by the total time in the nighttime period). The independent variables were the trial night (the specific night of the trial: 1–7), trial type (control trial, medium noise trial, or loud noise trial), trial night by trial type interaction, wind speed, precipitation, temperature, moon illumination, mass at the end of the trial, tarsus + metatarsus, and predator and non-predator presence. We included environmental and morphological variables within the model because these factors have been shown to impact antipredator behavior (e.g., wind speed: Carr & Lima, 2010; mass: Jones, Krebs & Whittingham, 2009). We performed a prior contrasts to compare specific trial nights.

The climate variables were obtained from a local weather station (http://iclimate.org; ACRE- West Lafayette). We calculated the mean of the wind speed (natural log transformed) and temperature across the nighttime period. Since there was no precipitation during 82% of trial nights, precipitation was categorized as being present or absent. Moon illumination was the fraction of the moon’s surface that was illuminated from the sun’s rays (http://www.timeanddate.com; Lafayette, IN). Predator and non-predator presence was whether predators or non-predators, respectively, were visible inside the cage or along the outside of the cage perimeter or not during the nighttime period (predators and non-predators were visible in only 50.8% and 68.5% of nights, respectively). We analyzed whether the amount of time that predators and non-predators spent near the experimental area (total time that predators or non-predators were visible during the nighttime period divided by the nighttime period) was related to trial type, trial night, trial type by trial night interaction, and environmental variables with a repeated-measures mixed linear model. We performed a prior contrasts to compare specific trial nights. We also performed a mixed linear model to assess whether the mass of the birds changed during the experiment; we calculated the percentage that the mass changed (mass on night 7 minus mass on night 1 divided by mass on night 1) and determined whether the trial type (control trial, medium noise trial, or loud noise trial) impacted this percentage.

We ran another two repeated-measures mixed linear models to determine the factors influencing when the birds ascended to the roost and descended from the roost for the night. The independent variables were the trial type (control trial, medium noise trial, or loud noise trial), trial night (the specific night of the trial: 1–7), trial type by trial night interaction, environmental variables during the nighttime period (wind speed, precipitation, temperature, and moon illumination), morphological measurements of the bird (mass and tarsus + metatarsus), and predator and non-predator presence. We performed binomial tests (Proc Freq) to assess peahens’ roosting preferences (the peahens did not switch to a different roost during a given night). We examined whether trial type (medium or loud noise), environmental variables (wind speed, temperature, and moon illumination), and morphological variables impacted roost choice using a binomial logistic regression (PROC Logistic). The wind speed and temperature at sunset during the night of the trial were used in the analysis; precipitation at sunset during the night of the trial was excluded from this analysis because precipitation was recorded in only 7.5% of trials. Analyses were performed in SAS (9.3; Cary, NC, USA) or Minitab (15.1; Minitab Inc., State College, PA, USA). The data supporting this article are available in Harvard Dataverse: http://dx.doi.org/10.7910/DVN/FFEZQC.

Table 1 The impact of trial type, trial night, environmental and morphological variables, and predator and non-predator presence on head movement rate, the amount of time predators and non-predators spent near the experimental area, the times at which the birds ascended to and descended from the roost, and roost selection.

F values (numerator degrees of freedom, denominator degrees of freedom) are displayed along with p-values for A–E; chi-square values (degrees of freedom) are displayed along with p-values for F.

	A: head movement rate	B: predators	C: non-predators	D: ascend roost	E: descend roost	F: roost selection	
Trial type	3.57 (2,25) 0.043	5.80 (2,27) 0.008	0.52 (2,27) 0.60	0.33 (2,25) 0.72	6.48 (2,25) 0.0054	1.39 (1) 0.24	
Trial night	0.55 (6,108) 0.77	0.87 (6,138) 0.52	1.85 (6,138) 0.095	0.43 (6,123) 0.86	0.57 (6,121) 0.75	–	
Trial type * trial night	0.24 (12,108) 0.99	2.58 (12,138) 0.0041	0.43 (12,138) 0.95	0.50 (12,123) 0.91	0.97 (12,121) 0.48	–	
Wind	0.03 (1,108) 0.85	7.21 (1,138) 0.0081	1.11 (1,138) 0.29	0.24 (1,123) 0.62	3.11 (1,121) 0.081	0.40 (1) 0.53	
Precipitation	0.77 (1,17) 0.39	0.15 (1,19) 0.70	1.08 (1,19) 0.31	0.06 (1,18) 0.81	2.14 (1,18) 0.16	–	
Temperature	3.04 (1,108) 0.084	15.49 (1,138) 0.0001	5.74 (1,138) 0.018	24.06 (1,123) <0.0001	0.79 (1,121) 0.38	1.37 (1) 0.24	
Moon illumination	0.23 (1,108) 0.63	1.31 (1,138) 0.25	0.04 (1,138) 0.85	0.53 (1,123) 0.47	2.09 (1,121) 0.15	0.12 (1) 0.73	
Mass	1.16 (1,25) 0.29	–	–	0.35 (1,25) 0.56	3.09 (1,25) 0.091	1.17 (1) 0.28	
Tarsus + metatarsus	3.42 (1,25) 0.076	–	–	2.80 (1,25) 0.11	12.15 (1,25) 0.0018	0.089 (1) 0.77	
Predator presence	0.39 (1,23) 0.54	–	–	0.82 (1,22) 0.37	1.89 (1,23) 0.18	–	
Non-predator presence	0.34 (1,14) 0.57	–	–	0.15 (1,15) 0.70	0.15 (1,14) 0.70	–	

Table 2 Specific contrasts were performed to compare treatment effects and time effects with respect to head movement rate (df = 108) and the amount of time that predators spent near the experimental area (df = 138).

Within the treatment effects, we examined whether the control and noise trials differed on night 1, nights 2–5 (averaged), and nights 6–7 (averaged). Within the time effects, we examined whether there were differences within the control or noise trials on night 1 compared to night 2–5 (averaged), nights 2–5 (averaged) compared to nights 6–7 (averaged), night 1 compared to night 6–7 (averaged), and night 2 compared to night 5. Contrasts were considered significant if they are less than the Bonferroni corrected p-value (18 contrasts; p < 0.0028).

			A: head movement rate	B: predators	
Treatment effects					
	Night 1	Control vs. medium noise	0.14 (0.89)	0.86 (0.39)	
	Night 1	Control vs. loud noise	0.39 (0.70)	0.58 (0.56)	
	Nights 2–5	Control vs. medium noise	2.82 (0.0056)	4.62 (<0.0001)	
	Nights 2–5	Control vs. loud noise	1.66 (0.099)	3.61 (0.0004)	
	Nights 6–7	Control vs. medium noise	0.93 (0.35)	0.47 (0.64)	
	Nights 6–7	Control vs. loud noise	0.60 (0.55)	1.45 (0.15)	
Time effects					
	Control	Night 1 vs. night 2–5	1.19 (0.23)	2.4 (0.018)	
	Medium	Night 1 vs. night 2–5	0.53 (0.60)	1.66 (0.10)	
	Loud	Night 1 vs. night 2–5	0.56 (0.58)	0.89 (0.38)	
	Control	Night 2–5 vs. night 6–7	0.02 (0.98)	1.92 (0.057)	
	Medium	Night 2–5 vs. night 6–7	1.44 (0.15)	1.15 (0.25)	
	Loud	Night 2–5 vs. night 6–7	0.54 (0.59)	2.98 (0.0034)	
	Control	Night 1 vs. night 6–7	1.04 (0.30)	0.71 (0.48)	
	Medium	Night 1 vs. night 6–7	0.53 (0.60)	0.65 (0.51)	
	Loud	Night 1 vs. night 6–7	0.90 (0.37)	1.40 (0.16)	
	Control	Night 2 vs. night 5	0.74 (0.46)	0.12 (0.91)	
	Medium	Night 2 vs. night 5	0.15 (0.88)	1.33 (0.19)	
	Loud	Night 2 vs. night 5	0.39 (0.70)	0.07 (0.94)	

Figure 1 Head movement rates (means ± SE) of peahens during noise (medium and loud) and control trials.

Figure 2 Amount of time that predators were present (means ± SE) during noise (medium and loud) and control trials.

Results

The head movement rate of peahens was similar regardless of trial night, the interaction between trial type and trial night, environmental variables (wind, precipitation, temperature, and moon illumination), morphological variables (mass and tarsus + metatarsus:), and predator presence and non-predator presence. However, the head movement rate of peahens was lower during noise trials compared to control trials (Table 1A). Comparing noise versus control trials on nights with and without noise, the head movement rates were similar (Table 2A: Treatment effects). There was a non-significant trend for head movement rate to be lower during nights with noise compared to nights without noise, especially during medium noise trials. Within the noise trials, the head movement rates were similar on nights with and without noise pollution; within the control trials, the head movement rates were similar across nights (Table 2A: Time effects; Fig. 1). The results were qualitatively similar when the medium and noise trials were pooled. There was no change in body mass within trials with respect to whether the birds were exposed to artificial noise pollution or not (F2,24 = 1.29, p = 0.29).

The amount of time that predators spent near the experimental area varied depending on trial type, the interaction between trial type and trial night, wind speed, and temperature but not trial night, precipitation, or moon illumination (Table 1B). Predators spent more time near the experimental area during control versus noise trials (Fig. 2), when the wind speed was low, and the temperature was high. They also spent more time near the experimental area during control trials compared to noise trials during nights when the noise was broadcast in noise trials (Table 2B: treatment effects). Within the noise trials and within the control trials, the amount of time that predators spent near the experimental area did not vary (Table 2B: time effects). The amount of time that non-predators were near the experimental area was only impacted by the temperature; the other variables were not significant (Table 1C). Non-predators spent more time near the experimental area when the temperature was high.

Peahens ascended to the roost later in the evening when the temperature was higher; the other independent variables did not affect when the birds ascended to the roost (Table 1D). Peahens descended from the roost later in the morning during control trials compared to noise trials (control: 24.3 ± 7.8 min after sunrise; medium noise: 16.9 ± 7.5 min after sunrise; loud noise: 12.2 ± 6.5 min after sunrise) and when their tarsus + metatarsus was longer; the other independent variables were not significant predictors of the time when the peahens descended from the roost (Table 1E). Peahens did not exhibit a preference for roosting closer or further from artificial noise (medium noise: 60% of the birds roosted away from the noise, p = 0.50; loud noise: 55% of the birds roosted away from the noise, p = 0.82; two-tailed binomial test). The type of noise (medium or loud), wind speed, temperature, moon illumination, mass, and tarsus + metatarsus did not impact whether the peahens roosted near or far from the noise (Table 1F).

Discussion

The nocturnal vigilance levels of peahens were not significantly impacted by noise pollution within trials. Individual peahens exhibited similar rates of head movements (a proxy of vigilance; Jones, Krebs & Whittingham, 2007) at night regardless of whether noise pollution was present or absent. Furthermore, they showed no preference for roosting away from artificial noise pollution.

The results suggest that peahens’ perception of risk is not drastically impacted by noise pollution. They did not increase their vigilance behavior to compensate for a potentially reduced ability to detect threats nor did they decrease their vigilance levels to take advantage of a potentially safer environment within trials. In most of the studies examining vigilance behavior and noise pollution, individuals elevate their vigilance levels in response to noise pollution (however, these studies were conducted on diurnal species during the day while this study was conducted at night; reviewed in Beauchamp, 2015): California ground squirrels (Otospermophilus beecheyi) are more vigilant in areas with turbine noise (Rabin, Coss & Owings, 2006), great tits (Parus major) are more vigilant when exposed to aircraft noise (Klett-Mindo, Pavón & Gil, 2016), prairie dogs (Cynomys ludovicianus) and white-crowned sparrows (Zonotrichia leucophrys) are more vigilant in response to traffic noise (Shannon et al., 2014a; Ware et al., 2015), chaffinches are more vigilant in response to white noise (Quinn et al., 2006), and koalas (Phascolarctos cinereus) are more vigilant when hearing zoo visitors (Larsen, Sherwen & Rault, 2014). This increased level of vigilance may allow animals to detect threats faster (Meillère, Brischoux & Angelier, 2015). However, some species may decrease their vigilance levels in response to noise pollution because their risk perception is lower. Elk are less vigilant in response to traffic noise (though human activity may have also contributed to this effect; Shannon et al., 2014b).

In addition, noise pollution did not influence the peahens’ selection of nocturnal roosting locations. The peahens selected roosts irrespective of noise pollution levels, indicating that they did not perceive noise pollution as impacting their risk. Because both roosts were exposed to some level of noise pollution, it is possible that the peahens did not distinguish between them since they were both noisy. Additional experiments in which one of the roosts is completely free of noise would be important. Previous studies have found that some species avoid areas with noise pollution (Blickley, Blackwood & Patricelli, 2012; Ware et al., 2015) while other species do not (Neo et al., 2015); an understanding of ecological differences between species could elucidate why they respond differentially.

Even though we did not find that vigilance levels differed within trials, we did find that peahens were more vigilant overall during control compared to noise trials. And, peahens descended from the roost later in the morning overall during control trials compared to noise trials. These overall effects could be related to predators being more frequent during control trials (see below) and peahens adjusting to this increase in predator presence throughout the trial (i.e., carry-over effects). In fact, head movement rates across all nights during control trials in this study (171 ± 9.9 head movements/h) were higher than control trials from a previous study conducted under similar experimental conditions (99 ± 6.5 head movements/h; Yorzinski et al., 2015); this suggests that changes in the external environment, such as increased predators, may have resulted in the higher vigilance rates during our control trials in this study. Because predator presence can impact vigilance behavior, future experiments could be conducted in which predator presence is controlled. Additional experiments will also be necessary to determine whether long-term effects of noise pollution impact vigilance behavior.

Peahens may not rely strongly on acoustic cues when detecting nocturnal predators, potentially explaining why they do not alter their perception of risk based on noise pollution within trials. Nocturnal vigilance levels in peahens dramatically increases with exposure to artificial light pollution (Yorzinski et al., 2015), suggesting that the birds heavily rely on vision to detect predators. Given that peahens in the wild roost atop tall trees, they also likely rely on vibrations to detect the approach of large predators. Noise pollution may mask acoustic cues from predators but be less important than visual or vibrational cues to the peahens. Given that their antipredator vocalizations are loud and cover a wide frequency range (Yorzinski, 2014), peahens may also be able to hear conspecific warning calls despite noise pollution (Francis, 2015; Pettinga, Kennedy & Proppe, 2016). Additional studies that examine whether peahens use acoustic cues during predator detection in the daytime would be useful. A comparative study examining variation in species’ response to noise pollution would help elucidate the factors impacting the perception of risk in response to noise pollution across species.

It is possible that the peahens in this study had previously habituated to noise pollution. The peahens were captured from feral populations located in country areas or suburban neighborhoods at least two years prior to the onset of this study. They may therefore have habituated to noise pollution while they were feral. However, the loudness and duration of the noise pollution they experienced while they were feral were likely less than the loud noise treatment in this study. It is also possible that the birds habituated to the noise during the trials because the noise was broadcast for four consecutive days. However, their nocturnal vigilance levels were similar on the first and last day when the noise was broadcast, suggesting that habituation within trials was not influencing vigilance rates.

Because nocturnal vigilance is inversely correlated with sleep in peahens (Yorzinski et al., 2015), noise pollution did not likely impact the amount of time the birds spent sleeping within trials. In addition, they ascended the roost in the evening and descended from the roost in the morning at similar times regardless of whether noise pollution was present or not within trials. However, noise pollution could have affected their sleep patterns in subtler ways. In humans, noise pollution can alter the amount of time spent in different stages of sleep (Pirrera, Valck & Cluydts, 2010). Further research investigating the impact of noise pollution on sleep in birds would be useful. One study found that European robins (Erithacus rubecula) sing at night when they are exposed to noise pollution but how this behavioral change impacted their sleep behavior was not explored (Fuller, Warren & Gaston, 2007).

In our study, predators often avoided the experimental area during nights with noise pollution. Similarly, Francis, Ortega & Cruz (2009) found that avian predators avoided depredating nests in areas exposed to noise pollution. In addition, three-spined sticklebacks (Gasterosteus aculeatus) and greater mouse-eared bats (Myotis myotis) are less efficient hunters when subjected to noise pollution (Purser & Radford, 2011; Siemers & Schaub, 2011). Because some predators may be sensitive to noise pollution, prey may be safer and have higher reproductive success in noisy environments (Francis, Ortega & Cruz, 2009). However, in other species, prey are more easily captured by predators when exposed to noise (Simpson, Purser & Radford, 2015; Simpson et al., 2015). The mechanisms underlying differences in hunting behavior across species in response to noise are not well understood.

Other types of anthropogenic disturbances, such as light and chemical pollution, often accompany sources of noise pollution (Halfwerk & Slabbekoorn, 2015). Understanding how different types of disturbances singly and jointly influence antipredator behavior would be informative. While we found that noise pollution had limited effects on nocturnal vigilance rates in peahens in this study, we previously found that light pollution significantly increases their nocturnal vigilance rates (similar sample size as used in this study; Yorzinski et al., 2015). Across nights with exposure to pollution, nocturnal vigilance rates in response to noise pollution in this study (medium noise: 118 ± 3.5 head movements/h; loud noise: 112 ± 6.0 head movements/h) were two times lower compared to vigilance rates in response to light pollution in a similar study (246 ± 38.6 head movements/h; Yorzinski et al., 2015). Based on the evidence thus far, management practices aimed at minimizing artificial disturbances to peafowl might therefore invest more in reducing light pollution compared to noise pollution.

We thank the Purdue Department of Forestry and Natural Resources for allowing us to house the birds on their property and providing logistical support. Barny Dunning also provided valuable logistical support. Jacie Anderson, Sarah Bischoff, Sydney Byerley, Jeanne Coy, Bridget Craft, Connor Egyhazi, Sara Green, Amanda Gnerlich, Diamond Jones, and Rachel Schultz helped performed the trials. Rebecca Budd, Bess Fary, Elissa Hall, Donica Owsley, and Brooke Staley analyzed some of the video recordings.

Additional Information and Declarations

Competing Interests

Author Contributions

Animal Ethics

Data Availability

The authors declare there are no competing interests.

Jessica L. Yorzinski conceived and designed the experiments, performed the experiments, analyzed the data, contributed reagents/materials/analysis tools, wrote the paper, prepared figures and/or tables, reviewed drafts of the paper.

Fredrick S. Hermann performed the experiments, reviewed drafts of the paper.

The following information was supplied relating to ethical approvals (i.e., approving body and any reference numbers):

The study was approved by Purdue University Animal Care and Use Committee (#1305000862 & 1504001232).

The following information was supplied regarding data availability:

Yorzinski, Jessica. 2016. Vigilance Levels. Harvard Dataverse, V1. DOI: 10.7910/DVN/FFEZQC.

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
