# Peer review of "Noise pollution has limited effects on nocturnal vigilance in peahens"

_PeerJ, doi:10.7717/peerj.2525_

## Round 0.1 · original submission · Major Revisions

We have received two excellent reviews of this manuscript. Both reviewers were positive about the general design, analysis and presentation of your research. However, they had some concerns especially regarding the statistical analysis, interpretation of results, and the ethics of the treatment noise level. In my reading, I also noted that your figure captions read more like statements of results than full descriptions of the figures. Also, you did not consistently provide scientific names at first use for species mention.

Reviewer 1 ·

Basic reporting

Good

Experimental design

Good

Validity of the findings

I have some doubts about the validity of some of the statistical models. These are explained below.

Additional comments

In this experiment, the authors tested whether peahens adjusted their nocturnal vigilance when exposed to increased noise levels. Several earlier studies suggested that noise pollution can have consequences for behaviour and this experiment allows us to see whether vigilance, one component of antipredator behaviour, can be influenced by noise pollution for prey animals at night. The results suggest that noise pollution had little impact on antipredator vigilance.
My main concern is that the statistical analysis was not optimal (see below). In any case, the direction of the effect, a reduction in vigilance on noisy nights, is intriguing as it is in the opposite direction to what we would predict. From the figure, noise seems to have little effect on vigilance; what we see is actually an increase in vigilance in the control trials on days 2-5. Any idea why we would get such as increase in vigilance on these days?
Line 55: Noise pollution can be just a change in noise level, but it is often associated with visual stimuli as well (think of a noisy road, for example, or noisy passersby). Perhaps it would be useful to make a distinction between the two types of situations early since this study only focuses on changes in noise levels without any associated visual stimuli.
Line 84: Would we expect a different type of reaction to noise pollution at these two periods? I am trying to justify why it would be important to investigate the effect of noise pollution at night. One might imagine that since visual vigilance is difficult to carry out at night, noise pollution might be more of an issue as it would disrupt auditory vigilance. Some justification along these lines would be welcome.
Line 128: Can you define SPL, please?
Line 203: You presumably mean the independent variables here I suppose.
Line 204: I am not sure I really agree with pooling the results from the two types of trials. Since it was part of the original design, I suggest leaving it to the reader to judge the effect.
Line 216: I think we need some justification for this analysis. Why would we expect to see a difference and why does it matter? Should this be part of the introduction?
Line 239: I do not understand why the denominator degrees of freedom are not the same for all the independent variables. Were they not tested in the same model?
Line 244: Are these contrasts performed using the mixed model or do they simply represent independent t-tests performed using data for each trial separately? This is not the same, and the first approach would be much preferable. I would also suggest using some type of correction for the alpha level since you are testing several hypotheses.
Line 249: I think better contrasts (using “estimate” in SAS terminology and “divisor” when using averages) would be the following:
To test time effects:
average of days 2-5 v. day 1 in control and treated groups
average of days 2-5 v. average of days 6 and 7 in control and treated groups
day 1 v. average of days 6 and 7 in control and treated groups
day 2 v. day 5 in control and treated groups (to test an habituation effect)
To test treatment effect
c v. t on day 1
c v. t using the average of days 2-5
c v. t using the average of days 6 and 7

Line 259: I suggest the same approach here as for the head movement rate analysis.
Line 285: Perhaps this conclusion will change with the new statistical analyses.
Line 295: A review of this literature is available in the following reference: Beauchamp G (2015) Animal vigilance: Monitoring predators and competitors. Academic Press, Oxford.
Line 316: But this would only work for terrestrial predators of quite a large size. We would need more evidence to rule out auditory vigilance. Perhaps trials conducted during the day would be needed.
Line 327: To help us judge the matter, can you justify the noise levels selected here in comparison to what they would normally experience? A simple explanation for the lack of effect is that the experimental noise level was too low or alternatively that there was a lack of statistical power to detect differences.

Reviewer 2 ·

Basic reporting

In general this is an interesting study addressing a topic not hitherto examined and with scientific merit.
- Generally well written and clear. Good logic between paragraphs, but there's a lack of transitions between sentences, particularly in the introduction (see pdf comments for more details).
- Figures are clear but Figure 3 doesn't add much and could be removed (put the details in the text).
- Raw data supplied for most of the analyses, but I could not find the data for times of ascent/descent to/from the roost.

Experimental design

Nice, explicit predictions outlining three separate hypotheses addressing questions that haven't yet been addressed. This study is well done and very detailed. Statistical modelling is complex due to many covariates and within treatment (among trial nights) and between treatment (between control and noisy groups). There is nothing dramatically wrong with this design, but an alternative design may have been easier to interpret (perhaps paired noise/control trials on a single bird, separated by a week or month).

However, I am concerned about the ethics of exposing birds to such high levels of noise and for such a long time. It isn't clear what decibel weighting was used to measure the noise exposure, but regardless, 75 dB, is louder than that used in most experiments conducted on birds (generally they range from 60 to 65 dB(A)), and 90 dB is extremely loud. I have only encountered one other study that used similar noise levels* and in that experiment birds were only exposed to each treatment for 3 minutes (3 min control, 3min at 70dB(A), 3min at 90dB(A)), and birds were unconfined. According to L125-126, noise in this study wasn't merely a night time treatment, it was a continuous treatment from noon Day 2 to noon Day 6; continuous exposure over 4 days is a long time. I think the manuscript requires a comment/reference outlining why this level of exposure (both time and amplitude) were necessary and/or acceptable to alleviate concerns regarding the ethics of this methodology.

* McLaughlin KE, Kunc HP. 2013. Experimentally increased noise levels change spatial and singing behaviour. Biology letters 9:20120771–20120771.

What is the rational for including so many covariates? Their potential effects are never mentioned.

Effect sizes (regression estimates) aren't reported, but this is a general limitation of Anova tables, and is a common practice. The authors state “more” and/or “less” but having an idea of how much more or less yields a better understanding of the biological significance (i.e. if the peahens descend from their roosts later in Control trials, how much later? 5min? 2 hours? These are very different effects).

Further, in some cases it appears that either different summaries are presented or a different statistical analysis was performed than is stated in the methods (e.g., comparing trial nights, L245-248, L249-250, L259-262, L263-264). If tables were used to report the findings it would give a clearer picture of how the analyses were performed (and would provide a place to insert regression estimates).

Validity of the findings

I was disappointed to see that the between treatment effects were omitted from the discussion. For example:

L243-244 “However, the head movement rate of peahens was lower during noise trials compared to control trials”
L284-285 “Peahens exhibited similar rates of head movements ... at night regardless of whether noise pollution was present or absent”

With four days of continuous noise, it is entirely possible that the vigilance patterns on nights 6 and 7 are experiencing carry-over effects from previous exposure to noise. Thus, only considering within treatment effects (among trial nights) may be result in missing some of the story.

The effects of noise on vigilance in peahens in this study may be difficult to interpret, but it is an oversimplification to simply state that there is no effect. I would suggest including a short discussion concerning why there may be differences between the treatment groups, but not within the noise treatment, among trial nights. Particularly because a reduction in vigilance in peahens couples very nicely with the finding that predators avoid the enclosure during noisy periods.

I would also amend the abstract to be less conclusive regarding the lack of effect, and I think that a discussion of the potential for habituation within a trial would be useful.

L275 “Peahens descended from the roost later in the morning during control trials compared to noise trials”
L332-334 “In addition, they ascended the roost in the evening and descended from the roost in the morning at similar times regardless of whether noise pollution was present or not.”

Once again, there is an overall treatment effect which is omitted from the discussion. If peahens descend later during control trials (and thus descended earlier during noise trials), it suggests there are consequences of noise on their sleep (L330-340)

In the roost selection experiment, with such loud noise levels was it possible that both roosts were deemed 'noisy' even if they experienced slightly different levels?

Additional comments

Please see the annotated pdf for more specific comments as well.

Annotated reviews are not available for download in order to protect the identity of reviewers who chose to remain anonymous.

---

## Round 0.2 · accepted · Accept

Both reviewers found the changes in your manuscript satisfactory and considered it suitable for publication. However, Reviewer 2 still had concerns about the sound amplitude, rejecting the manuscript only because he/she felt that this amplitude was unnecessary for the study. I agree that the manuscript is now ready for publication. Because the experiment was approved by an appropriate animal care committee and the issue of amplitude was addressed in the text, I am willing to accept the study.

Reviewer 1 ·

Basic reporting

Good

Experimental design

Good

Validity of the findings

Good

Additional comments

Thank you for taking my comments into account. I have only two minor comments. In table 2, it might be useful to indicate what the values in brackets stand for (p-value). The Bonferroni correction is very conservative. Next time, you might want to consider the Benjamini-Hochberg sequential procedure. It gives more power than the Bonferroni approach.

Reviewer 2 ·

Basic reporting

The authors have addressed my comments well. I have no further issues.

Experimental design

There are many contrasts but the tables do help to resolve the complexity.

With respect to the ethics, the authors have responded well to my concerns. However, I still feel that 90 dB(A) is extremely loud given 4 days of exposure, and I'm not convinced it was a requirement of the study.

I'm satisfied with all other changes the authors have made regarding the experimental design.

Validity of the findings

The authors have addressed all my concerns and I think the new discussion points are good additions.